# Reliability and Validity of Non-Instrumental Clinical Assessments for Adults with Oropharyngeal Dysphagia: A Systematic Review

**DOI:** 10.3390/jcm12020721

**Published:** 2023-01-16

**Authors:** Reinie Cordier, Renée Speyer, Matthew Martinez, Lauren Parsons

**Affiliations:** 1Department of Social Work, Education and Community Wellbeing, Northumbria University, Newcastle upon Tyne NE7 7XA, UK; 2Curtin School of Allied Health, Faculty of Health Sciences, Curtin University, Perth, WA 6102, Australia; 3Department of Health & Rehabilitation Sciences, Faculty of Health Sciences, University of Cape Town, Cape Town 7935, South Africa; 4Department Special Needs Education, Faculty of Educational Sciences, University of Oslo, 0318 Oslo, Norway; 5Department of Otorhinolaryngology and Head and Neck Surgery, Leiden University Medical Centre, 1233 XA Leiden, The Netherlands

**Keywords:** psychometrics, instrument development, measurement, deglutition, swallowing disorders, internal consistency, hypothesis testing, structural validity, content validity

## Abstract

This systematic review on non-instrumental clinical assessment in adult oropharyngeal dysphagia (OD) provides an overview of published measures with reported reliability and validity. In alignment with PRISMA, four databases (CINAHL, Embase, PsycINFO, and PubMed) were searched, resulting in a total of 16 measures and 32 psychometric studies included. The included measures assessed any aspect of swallowing, consisted of at least one specific subscale relating to swallowing, were developed by clinical observation, targeted adults, and were developed in English. The included psychometric studies focused on adults, reported on measures for OD-related conditions, described non-instrumental clinical assessments, reported on validity or reliability, and were published in English. Methodological quality was assessed using the standard quality assessment QualSyst. Most measures targeted only restricted subdomains within the conceptual framework of non-instrumental clinical assessments. Across the 16 measures, hypothesis testing and reliability were the most reported psychometrics, whilst structural validity and content validity were the least reported. Overall, data on the reliability and validity of the included measures proved incomplete and frequently did not meet current psychometric standards. Future research should focus on the development of comprehensive non-instrumental clinical assessments for adults with OD using contemporary psychometric research methods.

## 1. Introduction

Oropharyngeal dysphagia (OD) is a symptom or a collection of symptoms of one or more underlying anatomical abnormalities or impairments and disorders in cognitive, sensory, and motor acts involved with transferring food and liquids from the mouth to the stomach [1]. OD may result in reduced efficiency and safety of swallowing, failure to maintain hydration and nutrition, risk of choking and aspiration leading to pulmonary complications, and reduced quality of life [2]. Due to these serious sequelae compromising people’s health, dysphagia is one of the leading causes of death and morbidity for, but not limited to, older persons, children, and adults with neurological disorders (e.g., cerebral palsy, stroke, and dementia) and head and neck cancer patients [3]. To reduce the devastating effects of OD, early diagnosis and intervention are crucial in a patient’s illness trajectory.

The first step in the management of OD is screening to identify people at risk of dysphagia. Next, those patients who fail screening are referred for further clinical assessment, for example, to identify possible causes of the swallowing problems, estimate the efficacy and safety of swallowing including the risk of aspiration, support decisions on oral or alternative feeding routes, and establish baseline data for future reference when determining the effects of interventions or the impact of a disease over time [4]. Clinical assessment may involve either instrumental or non-instrumental assessment or both. As instrumental assessment (e.g., fiberoptic or videofluoroscopic evaluation of swallowing recordings) can diagnose aspiration, including silent aspiration and other physiological problems in the pharyngeal phase, instrumental assessment is often referred to as the ‘gold standard’ assessment. However, no international consensus exists about which visuoperceptual measure should be used for the evaluation of swallowing recordings, and access to instrumental assessment may not always be guaranteed due to its restricted availability [5]. Moreover, the psychometric properties of many existing visuoperceptual measures are either unknown or lack methodological robustness in line with current psychometric standards [5].

Non-instrumental clinical assessment by dysphagia experts refers to an alternative method of evaluation after failed screening comprising a large variety of assessments, each of which may describe different aspects of OD given that it is a multidimensional phenomenon (e.g., medical history taking, conducting a physical examination, and patient-reported functional health status or dysphagia-related quality of life). In the literature, different combinations of non-instrumental clinical assessments can be found, typically including measures of cognition and communication; evaluation of the oral, laryngeal, and pharyngeal anatomy, physiology, and function (including cranial nerve examination); oral intake, nutritional status, and mealtime observations; and intervention trials (e.g., bolus modification, head and postural adjustments, and/or swallow manoeuvres) [4,6].

In 2022, the European Society for Swallowing Disorders (ESSD) published recommendations on how to select the best evidence-based screening and non-instrumental assessments for use in clinical practice targeting different constructs, subject populations, and respondents, based on criteria for diagnostic performance, psychometric properties (reliability, validity, and responsiveness), and feasibility [6]. The ESSD also emphasised discontinuing the use of non-validated dysphagia assessments and implementing measures that demonstrate robust psychometric properties. To date, several systematic reviews have been published summarising the diagnostic performance of screening tools (e.g., Benfield, Everton [7], Bours, Speyer [8], Brodsky, Suiter [9], Kertscher, Speyer [10], O’Horo, Rogus-Pulia [11]) and the psychometric properties of visuoperceptual measures to evaluate fiberoptic or videofluoroscopic swallowing recordings [5], patient self-reported functional health status and quality-of-life questionnaires [12,13], and pediatric clinical assessments [14]. To date, no psychometric overview of clinician-reported non-instrumental clinical assessments in adults with OD has been published.

The purpose of this systematic review was to (a) summarise the characteristics of the identified non-instrumental clinical assessments for adults with OD (excluding patient self-report), (b) determine which psychometric properties related to reliability and validity were reported, and (c) construct a conceptual map of the identified measures to determine how comprehensive existing non-instrumental clinical assessments are in measuring all the underlying constructs. The reporting on the psychometric properties of measures was based on the terminology and definitions as defined in the COSMIN (Consensus-based Standards for the Selection of health Measurement Instrument) taxonomy [15,16]. Responsiveness (i.e., the ability of an instrument to detect change over time) was outside the scope of this review.

## 2. Materials and Methods

This systematic review was conducted and reported according to the Preferred Reporting Items for Systematic reviews and Meta-Analyses (PRISMA) 2020 statement and checklist [17]. The PRISMA statement and checklist (Appendix A) aim to enhance the essential and transparent reporting of systematic reviews. To report on psychometrics, terminology and definitions as defined in the COSMIN taxonomy were used [15,16] (Appendix A).

### 2.1. Data Sources and Search Strategies

A systematic literature search was performed across four electronic databases: CINAHL, Embase, PsycINFO, and PubMed. All publication dates up to 14 February 2022 were included. Both subject headings and free text terms related to dysphagia, non-instrumental clinical assessment, and psychometrics were used to capture all relevant literature. Table 1 presents the search strategies used within this review, outlined for each database. Following the initial round of abstract selection, a further literature search was performed across the same four electronic databases using the names and acronyms of included measures to identify eligible psychometric studies. All publications up to 06 June 2022 were included.

### 2.2. Eligibility Criteria

The eligibility of individual measures was determined through the following inclusion and exclusion criteria: (1) Measures assessed any aspect of swallowing (including oral intake), with measures investigating eating disorders or Gastro-Esophageal Reflux Disease (GERD) excluded; (2) at least one specific subscale or a minimum of 50% of the total number of items of the measures related to swallowing; (3) measures were developed for assessment by clinical observation or eliciting clinical information by questionnaire, with all instrumental assessments, screening tools, and self-reporting questionnaires excluded; (4) measures targeted adults (i.e., 18 years old and above); and (5) measures needed to be developed originally in English, excluding translated versions of these measures.

Psychometric studies included in the systematic review met the following inclusion and exclusion criteria: (1) Studies focused on adult populations (18 years old and above); (2) studies reported on measures for conditions related to OD or swallowing difficulties, whilst any studies related to psychogenic swallowing difficulties or eating disorders (e.g., anorexia or bulimia) were excluded; (3) studies described a non-instrumental clinical assessment, so any study that focused on instrumental assessment (e.g., videofluoroscopic or endoscopic evaluation of swallowing) was excluded; (4) studies reported on psychometrics—either validity or reliability—of the included measures as defined by the COSMIN taxonomy [15], thus excluding responsiveness; and (5) studies were published in English.

### 2.3. Abstract and Measure Selection

Two reviewers worked independently evaluating the abstracts and titles of the records returned from the initial database search against the eligibility criteria. Abstracts were reviewed separately by the two reviewers to ensure accuracy in study selection. Any disagreements between the reviewers were discussed and, where consensus could not be reached, a third reviewer was consulted to assist in finding a resolution. None of the three reviewers had any affiliations with any of the authors of the included studies or measures. The selection process was completed according to the PRISMA guidelines and flow diagram [17], thus no evident bias in article selection was present.

Following the initial database search, a further set of searches was performed including the names and acronyms of the included measures, with the aim of locating all eligible and relevant psychometric studies. The same procedure was followed to ensure the accuracy of the selection process. A separate search was undertaken to identify potential measures and studies that met the inclusion criteria from the reference lists of the included studies.

### 2.4. Data Extraction

Following the selection process of both the studies and the measures, data from the remaining articles were extracted using comprehensive data extraction forms. Data were extracted under the following categories: (1) Measure characteristics (e.g., purpose, target population, subscales, range of score) and (2) psychometric properties reported within the available studies. The use of a data extraction table ensured that the same data characteristics were extracted from all included papers [18]. One reviewer extracted all data, then a second reviewer checked the extracted data for accuracy.

### 2.5. Methodological Quality

The standard quality assessment (QualSyst), as described by Kmet et al., 2004 [19], was performed to evaluate the methodological strength and weaknesses of the included studies. The Qualsyst critical appraisal tool provides a systematic, reproducible, and quantitative means of evaluating the methodological quality of research over a broad range of study designs. Each of the 14 Qualsyst criteria is scored individually, whereafter a total score is converted to an overall quality percentage score (a total score divided by the number of applicable items and multiplied by 100). An overall quality percentage score of 80% or higher indicates strong methodological quality, a score between 70% and 79% indicates good quality, a score between 50% and 69% indicates adequate quality, and scores below 50% indicate poor methodological quality. The criteria for good psychometric properties were adapted from Prinsen and Mokkink [20]. All ratings were performed by two independent reviewers. After a consensus was reached, any studies with poor methodology ratings (<50%) were excluded.

### 2.6. Conceptual Mapping of Measures

To construct a conceptual map of the identified measures, this systematic review utilised OD non-instrumental clinical assessment theory and definitions to inform a deductive thematic analysis of the findings. Following thematic synthesis of the scales and subscales of the included measures by the first and second authors, domains, sub-domains, and elements were subsequently identified, resulting in a conceptual framework of non-instrumental clinical assessment of OD.

## 3. Results

### 3.1. Systematic Literature Search

From the initial search, 1430 records were retrieved from the four separate electronic databases: 277 from CINAHL, 579 from Embase, 40 from PsycINFO, and 534 from PubMed. Of these, 301 duplicates were removed. From the measure-specific search, 2513 records were retrieved from the four separate electronic databases: 312 from CINAHL, 1201 from Embase, 316 from PsycINFO, and 684 from PubMed. Of these, 478 duplicates were removed, leaving a combined 3164 articles to be reviewed. Figure 1 presents the flow chart of the studies and measures reviewed and excluded during the literature search according to the PRISMA [17]. Following this selection process, 377 studies were assessed for eligibility from which 141 individual measures were also assessed, leading to 345 studies as well as 125 measures being excluded (see Appendix A). Altogether, a total of 32 original psychometric studies that focused on OD or other swallowing difficulties and included a clinical non-instrumental measure for an adult population and 16 individual measures were included.

### 3.2. Characteristics of Included Measures and Psychometric Studies

Descriptions and characteristics of the included measures are presented in Table 2. All 16 included measures were either developed or adapted for adult populations, with seven (44%) developed for stroke patients [21,22,23,24,25,26,27], two (12.5%) developed for adults with intellectual disability [28], and 2 (12.5%) measures developed for patients with head and neck cancers [29,30]. Measures ranged from one single scale to five subscales, with item numbers ranging from 1 [31] to 42 [32]. All measures were developed for clinical use, whilst the Eating and Drinking Ability Classification System (EDACS) can also be administered by a caregiver [33].

### 3.3. Conceptual Mapping of Non-Instrumental Clinical Measures

The systematic review utilised OD non-instrumental clinical assessment theory and definitions to inform a deductive thematic analysis of the findings [6]. Based on the thematic analysis, three domains were first identified, followed by sub-domains that were identified and subsumed under the most relevant domain, followed by elements that were subsumed under the most relevant sub-domain. The purpose of the conceptual mapping was to analyse the included measures in relation to how comprehensively they assess the construct of non-instrumental clinical measurement.

The content of the included measures—subscales and their items—varied and covered three domains (Figure 2): (1) Skills Related to Eating and Drinking; (2) Making Adjustments to Facilitate Eating and Drinking; and (3) Swallowing Act. The first domain ‘Skills Related to Eating and Drinking’ consists of three subdomains: Eating skills (two elements: Self-feeding skills (e.g., setting up tray, grasping utensils, bringing food to mouth) and oral preparation (e.g., open mouth anticipation of food, stripping spoon, biting off, taking appropriate bolus size, sipping from cup, mastication)), oral motor skills (three elements: Movement and coordination, strength, and symmetry (e.g., of lips, tongue, soft palate)), and cognitive skills and sensory perception (two elements: Cognitive skills (e.g., alertness, cooperation, comprehension) and sensory perception (e.g., taste, smell)).

The second domain ‘Making Adjustments to Facilitate Eating and Drinking’ includes two subdomains: Modified aspects related to the environment (three elements: Instrumental feeding adaptation (e.g., adaptive utensils), adjustment of food and drink intake (e.g., bolus modification/food texture and drink consistency, caloric intake, nutritional supplements), and feeding support (e.g., cueing, prompting, adaptive swallowing strategies, guidance)), and modified aspects related to a person (two elements: Posture and head control (e.g., symmetrical upright sitting posture, supported head control, and alternative methods of feeding (e.g., dependency versus independency of the method of food intake: Non-oral, tube, or PEG versus oral intake)).

The third domain ‘Swallowing Act’ refers to two subdomains: Safety of swallowing and efficiency of swallowing. Safety of swallowing includes four elements (respiration (e.g., sputum upper airways, coordination of breathing and swallowing, pneumonia, chest status), pain and discomfort (e.g., globus feeling), pharyngeal or laryngeal clearance (e.g., aspiration, cough, choke, throat clearing, gag, pharyngeal response, laryngeal movement, voice change), and trache (i.e., tracheostomy or tracheostomy tube). The efficacy of swallowing consists of three elements (oral residue (e.g., oral food remains, multiple swallows to clear bolus, spitting food or drinks, sputum), speed (e.g., duration of completing meal, speed of oral intake, tiring), and direction (e.g., drooling or lip closure, regurgitation, vomiting, rumination)).

### 3.4. Validity Evidence

The validity properties of the measures—content validity, criterion validity (where applicable), and construct validity (i.e., hypothesis testing, structural validity, and cross-cultural validity (where applicable))—are summarised in Table 3. Additionally, Table 4 provides an overview of the psychometric properties reported for each measure.

Content validity was reported for 11 of 16 (69%) of the measures: DSRS [26], EDACS [33], EDSQ [35], FOIS [23], IDDSI-FDS [36], MASA [27], MASA-C [29], MISA [37], mMASA [22], Swallowing Status [39], and TOMASS [40]. Relevance was evaluated for three of these six measures (DSRS, IDDSI-FDS, and Swallowing Status) based on feedback from experts, either through a Delphi, survey, panel, or focus group. As no international agreement exists about gold-standard assessments in non-instrumental assessment in dysphagia, criterion validity—the degree to which the scores of an instrument are an adequate reflection of a “gold standard” [15,16]—could only be determined for the MASA-C [29] and the mMASA [22], comparing both measures with the original MASA [27].

The most commonly reported aspect of construct validity reported within the included studies was hypothesis testing—the degree to which the results produced evidence that was consistent with hypotheses based on the assumption that the instrument validly measures the construct to be measured [15,16]—with relevant data available for all 16 measures (see Table 4). Conversely, structural validity—the extent to which an instrument’s scores adequately reflect the dimensionality of the construct to be measured [15,16]—was reported for only three (19%) of the sixteen measures: DDS [28], MASA-C [29], and Swallowing Status [39]. Cross-cultural validity refers to the degree to which the performance of the items on a translated or culturally adapted instrument is an adequate reflection of the performance of the items of the original version of the instrument [15,16]. As translated versions of the included measures were excluded from this review, only other forms of measurement invariance as a parameter of cross-cultural validity were considered, if applicable. For two measures (MASA-C [29] and mMASA [22]), measurement invariance could have been determined but was not reported.

### 3.5. Reliability Evidence

The reliability domain properties of the measures—internal consistency, reliability (i.e., test/retest and intra/inter-rater agreement), and measurement error—are outlined in Table 3. Internal consistency was reported for nine of the sixteen (56%) measures and was calculated using Cronbach’s alpha in each case, with values ranging from “good” (α = 0.71) to “excellent” (α = 0.99), thus showing sufficient overall consistency for each of these measures [59].

Data on the reliability measurement property were reported for all but two measures. Inter-rater agreement was determined for 12 of the 16 (75%) measures, with Intraclass Correlation Coefficient (ICC) and Kappa coefficient being the most commonly reported. ICC was reported for eight measures (50%), with values ranging from “moderate” (ICC = 0.68) to “excellent” (ICC = 1.00) [60]. The Kappa coefficient was reported for five measures (31%), with values ranging from “moderate” (κ = 0.45) to “very good” (κ = 0.91) [61]. Additionally, intra-rater agreement was reported for four of the sixteen (25%) measures using ICC for all four measures, with all “excellent” values ranging from ICC = 0.94 to ICC = 1.00. Test–retest reliability was reported for four measures (25%), again all reported on using ICC, with values ranging from “moderate” (ICC = 0.571) to “excellent” (ICC = 1.00). For these four measures, the time interval between trials for test–retest reliability varied from approximately 15 min to six weeks. No data on measurement error were reported for any of the measures. Table 4 provides an overview of the reported psychometric properties within the domains of reliability and validity per measure.

### 3.6. Conceptual Mapping of Included Measures

Three assessment domains were identified: ‘*Skills Related to Eating and Drinking*’, ‘*Making Adjustments to Facilitate Eating and Drinking*’, and ‘*Swallowing Act*’ (see Figure 2). These three domains were separated into individual sub-domains and elements to help analyse the 16 non-instrumental clinical measures within this study.

Eight measures included items specific to the first domain of ‘*Skills Related to Eating and Drinking*’, with MASA [27] and MASA-C [29] including six of seven elements (all but ‘self-feeding skills) from this domain and EDACS [33] and TOMASS [40] each only including one element. All but one measure (TOMASS [40]) included items specific to the second domain of ‘*Making Adjustments to Facilitate Eating and Drinking*’, with DDS [28] and DMSS [28] including all five elements whilst seven measures—EDSQ [35], IDDSI-FDS [36], MASA [27], MASA-C [29], M-MASA [22], Swallowing Status [39], and SPEAD [30]—included only one element. Thirteen of the measures (81.3%) included items specific to the third domain of ‘*Swallowing Act*’, with EDACS [33] including six of the seven elements (all but Trache) from this domain whilst SPEAD [30] and TOMASS [40] each only including one element (Speed). Of the elements in the first domain, ‘Skills related to eating and drinking’, ‘Oral preparation’ was included the most (seven of sixteen) and ‘Sensory perception’ was included the least (two of sixteen). Of the elements in the second domain, ‘Making adjustments to facilitate eating and drinking’, ‘Adjust food & drink intake’ was included the most (13 of 16) and ‘Feeding adaptation’ was included the least (2 of 16). Finally, of the elements in the third domain, ‘Swallowing act’, ‘Pharyngeal or laryngeal clearance’ was included the most (10 of 16) and ‘Trache’ was included the least (2 of 16).

Overall, only seven of the sixteen measures included at least one item specific to each of the three domains, though thirteen of the sixteen included at least one item specific to two of the three domains. Three measures targeted a single domain only. The mean number of elements per measure was six (MN = 6.4; SD = 2.8). MASA [27] and MASA-C [29] included the most elements (12 of 19), whilst IDDSI-FDS [36] included the least elements (1 of 19) across the three domains.

### 3.7. Methodological Quality

Appendix A shows the outcomes of the QualSyst critical appraisal tool by Kmet et al. [19]. As all studies had sufficient methodological quality, no studies were excluded. The overall methodological quality was strong, with the 32 included studies ranging from 90–100% ratings across the ten aspects assessed. The item that was most commonly given either a “Partial” or “No” rating was item 10 “Analytic methods described/ justified and appropriate”, which resulted from not meeting the criteria for good psychometric properties.

## 4. Discussion

This systematic review, in line with the PRISMA guidelines [17], aimed to provide an overview of the psychometric properties of clinician-reported non-instrumental assessment in OD. A total of 16 measures were retrieved with published data on one or more psychometric properties within the validity and/or reliability domains. No data were available on measurement error and only three measures provided data on structural validity. As a result, none of the included measures provided a complete overview of its psychometric properties. Furthermore, data on validity and reliability as retrieved from the literature may not always meet current psychometric standards. In other words, measures providing data on its psychometric properties may not always meet methodological quality criteria as, for example, defined by COSMIN to support their implementation in research and daily clinical practice.

An important finding is that very few measures were identified in the literature that comprehensively measure the construct of non-instrumental clinical assessment, with seven measures (43.8%) consisting of a single scale covering only single aspects of OD. Based on the conceptual framework of non-instrumental clinical measures as introduced in this review, the identified measures collectively demonstrated great variety across a number of domains, subdomains, and elements. The number of elements for each measure ranged between one, demonstrating a very narrow focus, and twelve, demonstrating a very broad focus. On average, measures consisted of six elements. Even measures targeting all three domains as defined in our conceptual framework—skills related to eating and drinking, making adjustments to facilitate eating and drinking, and swallowing act—would still exclude several subdomains and elements from the assessment of people with OD. Therefore, since OD is a multidimensional phenomenon [6] and most measures only focus on restricted aspects of OD, clinicians should include multiple measures if aiming to capture the full concept of OD.

The conceptual framework also highlighted the importance of a multidisciplinary approach in the assessment of OD. Different professional healthcare workers may add value to ensure comprehensive evaluation across OD assessment domains. By combining expertise from different disciplines (e.g., speech pathologists, occupational therapists, nurses, psychologists, pulmonologists), OD can be evaluated in all its multidimensional aspects. Consequently, experts from all relevant disciplines should be involved at the onset of instrument development to ensure good content validity [15].

This current review is a first step towards optimising non-instrumental clinical assessment of OD. Although this review was based on the terminology and definitions used in the COSMIN taxonomy [15,16], it is recommended to conduct another, more in-depth psychometric review following the robust COSMIN methodologies and comparing psychometric data and statistical methods using quality criteria as formulated by the COSMIN group. This enables the quality of the psychometric studies and the quality of the psychometric properties to be thoroughly evaluated.

Future research should focus on developing more comprehensive non-instrumental clinical assessments that can be used to capture OD as a multidimensional phenomenon, using contemporary psychometric standards and methods such as item response theory and classic test theory. All psychometric properties should be determined and reported on to allow for validity, reliability, and responsiveness to be established. Finally, before implementing newly developed measures in research and clinical practice, feasibility aspects should be taken into consideration such as time constraints and accessibility. Non-instrumental clinical measures with robust psychometric properties could be of critical value, especially in those health settings where access to instrumental assessment is not possible or where availability is restricted.

Although the reporting of this review followed the PRISMA guidelines to reduce bias, some limitations are inherent to this study. As only studies and measures published in English were included, some measures may have been excluded based on language criteria. According to the COSMIN framework [15,16], nine psychometric properties should be considered if applicable. However, since no international consensus exists about a gold-standard non-instrumental clinician-reported assessment in OD, criterion validity was limited to comparisons between original measures and their revised versions (e.g., shortened versions or versions adapted to specific target populations). Furthermore, since only measures developed in English were included, translated versions of measures were excluded from this review, limiting cross-cultural validity to other forms of measurement invariance, such as different clinical populations. Further, as the identification of studies on responsiveness would have required different search strategies in the electronic databases, this psychometric domain was outside the scope of the current review.

## 5. Conclusions

This systematic review following PRISMA guidelines and terminology as defined by the COSMIN framework summarised the reliability and validity of non-instrumental clinical assessments for adults with OD excluding patient self-report. Only 16 measures were identified with reported psychometric characteristics. Even though OD is considered a multidimensional phenomenon, most measures only captured restricted subdomains within the conceptual framework of non-instrumental clinical assessments. Further, data on the reliability and validity of included measures proved incomplete and did not always meet current psychometric standards. Future research should focus on the development of comprehensive non-instrumental clinical assessments for adults with OD using contemporary psychometric research methods.

## Figures and Tables

**Figure 1 jcm-12-00721-f001:**
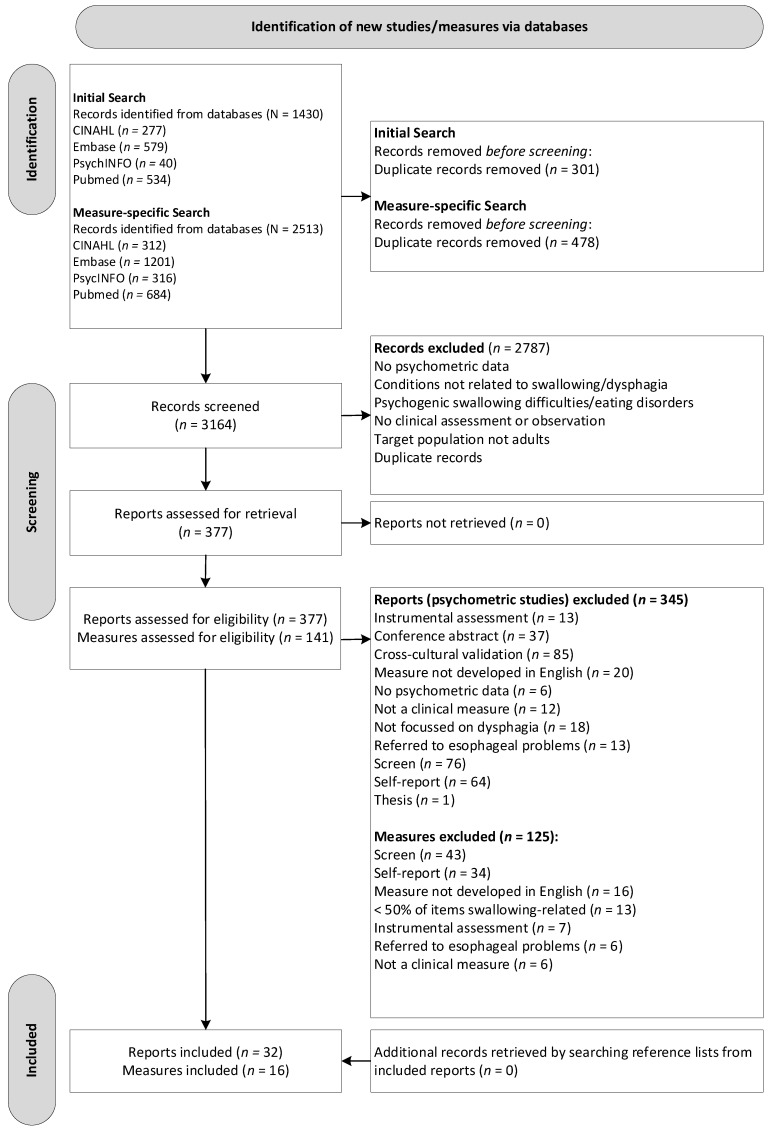
Flow diagram of the review process based on PRISMA.

**Figure 2 jcm-12-00721-f002:**
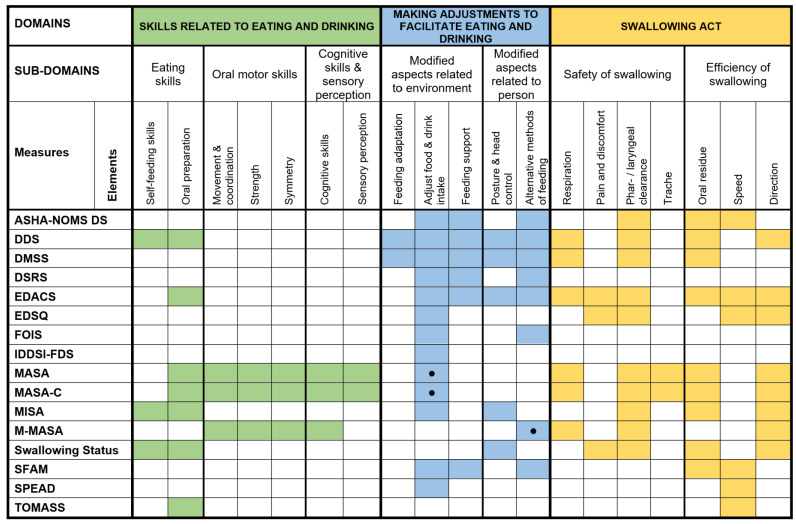
Conceptual mapping of non-instrumental clinical measure items. Note. Legend background colours. Green = Skills related to eating and drinking; Blue = Making adjustment to facilitate eating and drinking; Yellow = Swallowing act. *●* = Item, but not part of scoring; ASHA-NOMS DS = ASHA-NOMS Dysphagia Scale; DDS = Dysphagia Disorders Survey; DMSS = Dysphagia Management Staging Scale; DSRS = Dysphagia Severity Rating Scale; EDACS = Eating and Drinking Ability Classification System; EDSQ = Easy Dysphagia Symptom Questionnaire; FOIS = Functional Oral Intake Scale; IDDSI-FDS = International Dysphagia Diet Standardisation Initiative Functional Diet Scale; MASA = Mann Assessment of Swallowing Ability; MASA-C = Mann Assessment of Wallowing Ability-Cancer; MISA = McGill Ingestive Skills Assessment; M-MASA/mMASA = Modified Mann Assessment of Swallowing Ability; SPEAD = Swallowing Proficiency for Eating and Drinking; TOMASS = Test of Masticating and Swallowing Solids.

**Table 1 jcm-12-00721-t001:** Search terms per database.

Literature Database	Search Strategies
Cinahl Plus with Full Text (EBSCO)	((MH “Deglutition”) OR (MH “Deglutition Disorders”)) AND (ti (assess* OR index* OR indices OR instrument* OR measure* OR subscale* OR scale* OR screen* OR tool* OR survey* OR inventor* OR protocol* self-report* OR patient-report* OR observation* OR rating* OR rated OR score*) OR ti (clinical N2 (outcome* OR evaluat*)) OR ti (funct* N2 (outcome* OR evaluat*)) OR ti (quality of life N2 (outcome* OR evaluat*)) OR ti (health N2 (outcome* OR evaluat*))) AND ((MH “Psychometrics”) OR (MH “Measurement Issues and Assessments”) OR (MH “Validity”) OR (MH “Predictive Validity”) OR (MH “Reliability and Validity”) OR (MH “Internal Validity”) OR (MH “Face Validity”) OR (MH “External Validity”) OR (MH “Discriminant Validity”) OR (MH “Criterion-Related Validity”) OR (MH “Consensual Validity”) OR (MH “Concurrent Validity”) OR (MH “Qualitative Validity”) OR (MH “Construct Validity”) OR (MH “Content Validity”) OR (MH “Instrument Validation”) OR (MH “Validation Studies”) OR (MH “Test-Retest Reliability”) OR (MH “Sensitivity and Specificity”) OR (MH “Reproducibility of Results”) OR (MH “Reliability”) OR (MH “Intrarater Reliability”) OR (MH “Interrater Reliability”) OR (MH “Measurement Error”) OR (MH “Bias (Research)”) OR (MH “Selection Bias”) OR (MH “Sampling Bias”) OR (MH “Precision”) OR (MH “Sample Size Determination”) OR (MH “Repeated Measures”) OR Psychometric* OR reliabilit* OR validit* OR reproducibility OR bias) Narrow by Language: english Narrow by SubjectAge: all adult
Embase (Ovid)	(swallowing/OR dysphagia/) AND ((assess* OR index* OR indices OR instrument* OR measure* OR subscale* OR scale* OR screen* OR tool* OR survey* OR inventor* OR protocol* self-report* OR patient-report* OR observation* OR rating* OR rated OR score*).ti. OR (clinical adj2 (outcome* OR evaluat*)).ti. OR (funct* adj2 (outcome* OR evaluat*)).ti. OR (quality of life adj2 (outcome* OR evaluat*)).ti. OR (health adj2 (outcome* OR evaluat*)).ti.) AND (psychometry/OR validity/OR reliability/OR measurement error/OR measurement precision/OR measurement repeatability/OR error/OR statistical bias/OR test retest reliability/OR intrarater reliability/OR interrater reliability/OR accuracy/OR criterion validity/OR internal validity/OR face validity/OR external validity/OR discriminant validity/OR concurrent validity/OR qualitative validity/OR construct validity/OR content validity/OR Psychometric* OR reliabilit* OR validit* OR reproducibility OR bias) Limit to (adults <18 to 64 years> or aged <65+ years> Limit to english language
PsychINFO (Ovid)	(swallowing/OR dysphagia/) AND ((assess OR index* OR indices OR instrument* OR measure* OR subscale* OR scale* OR screen* OR tool* OR survey* OR inventor* OR protocol* self-report* OR patient-report* OR observation* OR rating* OR rated OR score*).ti. OR (clinical adj2 (outcome* OR evaluat*)).ti. OR (funct* adj2 (outcome* OR evaluat*)).ti. OR (quality of life adj2 (outcome* OR evaluat*)).ti. OR (health adj2 (outcome* OR evaluat*)).ti.) AND (Psychometrics/OR Statistical Validity/OR Test Validity/OR Statistical Reliability/OR Test Reliability/OR Error of Measurement/OR Errors/OR Response Bias/OR Interrater Reliability/OR Repeated Measures/OR Psychometric* OR reliabilit* OR validit* OR reproducibility OR bias) Limit to “300 adulthood <age 18 yrs and older>” Limit to english language
PubMed	(“Deglutition”[Mesh] OR “Deglutition Disorders”[Mesh]) AND (assess*[Title] OR index*[Title] OR indices[Title] OR instrument*[Title] OR measure*[Title] OR subscale*[Title] OR scale*[Title] OR screen*[Title] OR tool*[Title] OR survey*[Title] OR inventor*[Title] OR protocol* self-report*[Title] OR patient-report*[Title] OR observation*[Title] OR rating*[Title] OR rated[Title] OR score*[Title] OR (clinical[Title] AND (outcome*[Title] OR evaluat*[Title])) OR (funct*[Title] AND (outcome*[Title] OR evaluat*[Title])) OR (“quality of life” [Title] AND (outcome*[Title] OR evaluat*[Title])) OR (health[Title] AND (outcome*[Title] OR evaluat*[Title]))) AND (“Psychometrics”[Mesh] OR “Reproducibility of Results”[Mesh] OR “Validation Studies as Topic”[Mesh] OR “Bias”[Mesh] OR “Observer Variation”[Mesh] OR “Selection Bias”[Mesh] OR “Diagnostic Errors”[Mesh] OR “Dimensional Measurement Accuracy”[Mesh] OR “Predictive Value of Tests”[Mesh] OR “Discriminant Analysis”[Mesh] OR psychometric* OR reliabilit* OR validit* OR reproducibilit* OR bias) Adult: 19+ years, English

**Table 2 jcm-12-00721-t002:** Characteristics of the included measures.

Measurement Tool (Reference)	Purpose	Target Population	Measurement Type	Main Constructs Subscales (Number of Items per Subscale)	Response Options	Range of Score; Interpretation *
**ASHA-NOMS Dysphagia Scale** American Speech-Language-Hearing Association National Outcomes Measurement System Dysphagia Scale [34] Dungan, Gregorio [21]	A tool designed to measure both the supervision level required and diet level by assigning a single number that describes whether there has been a change in functional status after the speech therapy of patients with dysphagia	Stroke patients and those with brain lesions	Clinician-observed activities	**Single scale (1)—Levels:** Individual is not able to swallow anything safely by mouth. Individual is not able to swallow safely by mouth for nutrition and hydration Alternative method of feeding required as individual takes less than 50% of nutrition and hydration by mouth Swallowing is safe, but usually requires moderate cues to use compensatory strategies Swallowing is safe with minimal diet restriction and/or occasionally requires minimal cueing to use compensatory strategies Swallowing is safe, and the individual eats and drinks independently and may rarely require minimal cueing The individual’s ability to eat independently is not limited by swallow function	Single 7-level ordinal scale (1 = nothing by mouth; 7 = no limit by swallowing)	**Range:** 1–7 **Interpretation:** ↓ scores = ↑ dysphagia severity
**DDS** Dysphagia Disorders Survey Sheppard, Hochman [28]	A quantitative observation tool with capability for discriminating swallowing and feeding pathology from functionally competent patterns and providing an objective description of the clinical presentation of swallowing and feeding disorder in developmental disability (SFD-DD)	Adults and children with developmental disability	Clinician-observed	**Subscales (*n* _Item Total_ = 15):** **1.** **Related factors (RF; 7)** Body Mass Index (BMI) Independence Body postural control Diet consistency Adaptive utensils Special feeding techniques Seating supports/alignments **2.** **Feeding and Swallowing Competency (FSC; 8)** Orienting Reception Containment Oral transport Chewing Oral-pharyngeal swallow Post swallow Oesophageal swallow	Part 1: Rate progressive severity on each item (0–4) Part 2: Binary scoring for each item (0 = functionally correct; 1 = functionally deficient)	**Range:** Raw score (0–39; total from scores per subscale: RF = 0–17, FSC = 0–22) converted to DDS Percentile **Interpretation:** ↑ scores = ↑ dysphagia severity
**DMSS** Dysphagia Management Staging Scale Sheppard, Hochman [28]	An ordinal scale for presence and severity of swallowing and feeding disorder in developmental disability (SFD-DD)	Adults and children with developmental disability	Clinician-observed	**Single scale (1)—Levels:** No disorder Mild disorder Moderate disorder Severe disorder Profound disorder	Single 5-level ordinal scale (1 = no disorder; 5 = profound disorder)	**Range:** 1–5 **Interpretation:** ↑ scores = ↑ dysphagia severity
**DSRS** Dysphagia Severity Rating Scale Everton, Benfield [26]	Grades clinical dysphagia severity by quantifying how much modification is required to fluids and diet, as well as level of supervision, for safe oral intake	Stroke patients	Clinician-observed	**Subscales (*n* _Item Total_ = 3):** Fluid intake (1) Dietary intake (1) Supervision (1)	5-level ordinal assessment (0 = normal; 4 = no oral intake)	**Range:** 0–12 (total from scores per subscale) **Interpretation:** ↑ scores = ↑ dysphagia severity
**EDACS** Eating and Drinking Ability Classification System Sellers, Mandy [33]	A classification system for documenting and reporting how individuals with Cerebral Palsy (CP) eat and drink in everyday life.	Patients with CP	Clinician-observed (Optional: Caregiver-observed)	**Single scale (1)—Levels:** Eats and drinks safely and efficiently Eats and drinks safely but with some limitations to efficiency Eats and drinks with some limitations to safety; there may be limitations to efficiency Eats and drinks with significant limitations to safety Unable to eat and drink safely—tube feeding may be considered to provide nutrition	Single 5-level ordinal classification system (1 = eats and drinks safely and efficiently; 5 = Unable to eat and drink safely)	**Range:** 0–5 **Interpretation:** ↑ scores = ↑ dysphagia severity
**EDSQ** Easy Dysphagia Symptom Questionnaire Uhm, Kim [35]	A simple and rapid dysphagia questionnaire for older adults	Older adults (over 65 years old)	Clinician eliciting information based on questioning	**Single scale (12):** Do you have difficulty when you eat food or drink water? Do you have difficulty when you swallow a pill? Do you cough when you eat food or drink water? Do you choke when you eat food or drink water? Do you have feeling of something stuck in the throat when you swallow? Do you feel pain when you swallow? Do you take more than 30 min to eat an average meal? Do you have drooling or spitting out food during a meal? Have you ever been diagnosed with pneumonia? Have you lost weight due to swallowing difficulty? Do you have hoarse or wet voice after swallow? Do you get sputum after a meal?	Binary scoring for each item (Yes / No)	**Range:** 0–12 (total score from the sum of all “yes” responses) **Interpretation:** ↑ scores = ↑ dysphagia severity
**FOIS** Functional Oral Intake Scale Crary, Carnaby Mann [23]	To determine patients’ oral intake status, developed as an appropriate tool for estimating and documenting changes in the functional eating abilities of stroke patients over time	Stroke patients	Clinician-observed	**Single scale (1):** Nothing by mouth. Tube dependent with minimal attempts of food or liquid. Tube dependent with consistent oral intake of food or liquid. Total oral diet of a single consistency. Total oral diet with multiple consistencies, but requiring special preparation or compensations. Total oral diet with multiple consistencies without special preparation, but with specific food limitations. Total oral diet with no restrictions.	Single 7-level ordinal scale (1 = nothing by mouth; 7 = total oral diet with no restrictions)	**Range:** 1–7 **Interpretation:** ↑ scores = ↓ impairment severity
**IDDSI-FDS** International Dysphagia Diet Standardisation Initiative Functional Diet Scale Steele, Namasivayam-MacDonald [36]	To capture the severity of oropharyngeal dysphagia, as represented by the degree of diet texture restriction recommended for the patient	Patients with dysphagia risk	Clinician-observed	**Subscales:** Food level Drink level	Foods: 7—Regular 6—Soft & bite-sized 5—Minced & moist 4—Pureed (=Drinks 4) 3—Liquidised (=Drinks 3) Drinks: 4—Extremely thick (=Foods 4) 3—Moderately thick (=Foods 3) 2—Mildly thick 1—Slightly thick 0—Thin	**Range:** 0 (nothing by mouth)–8 (absence of diet texture restrictions); IDDSI-FDS score based on scoring chart (number in intersecting cell of food level column and drink level row) **Interpretation:** ↑ scores = ↓ impairment severity
**MASA** Mann Assessment of Swallowing Ability Mann [27]	Developed as a comprehensive clinical examination for identifying eating and swallowing disorders in patients with stroke.	Stroke patients	Clinician-observed	**Subscales (*n* _Item Total_ = 24):** General patient examination (6) Oral preparation phase (8) Oral phase (4) Pharyngeal phase (6)	3, 4 and 5-level ordinal scales (different weighting)	**Range:** Raw score (range 38–200; total score of all items) converted to severity grouping (no abnormality detected; mild; moderate; severe) for dysphagia and aspiration. **Interpretation:** ↑ scores = ↓ impairment severity
**MASA-C** Mann Assessment of Swallowing Ability—Cancer Carnaby and Crary [29]	Modified version of the MASA designed for cancer patients.	Patients receiving radiotherapy for head and neck cancer	Clinician-observed	**Adapted from MASA with subscales, but subscales undetermined for adapted measure (*n* _Item Total_ = 24):** Includes 15 of the original 24 items from the MASA, with an additional 9 cancer-specific items added	3, 4 and 5-level ordinal scales (different weighting)	**Range:** Raw score (range 40–200; total score of all items) converted to severity grouping (no abnormality detected; mild; moderate; severe) for dysphagia and aspiration. **Interpretation:** ↑ scores = ↓ impairment severity
**MISA** McGill Ingestive Skills Assessment Lambert, Gisel [37]	An evaluative tool that assigns a numerical score to the functional abilities of the patient in the domains of self-feeding, positioning, oral motor skills for solid and liquid ingestion, and overall feeding safety.	Elderly persons with neurologic impairments	Clinician-observed	**Subscales (*n* _Item Total_ = 42):** Positioning for meals (4) Self-feeding skills (7) Oral motor skills for solid consumption (12) Oral motor skills for liquid consumption (7) Texture management (12)	4-level ordinal scale (1–4)	**Range:** 42–126 (total from scores per subscale) **Interpretation:** ↑ scores = ↑ function
**M-MASA/** **mMASA** Modified Mann Assessment of Swallowing Ability Antonios, Carnaby-Mann [22]	Simplified version of the MASA designed to utilise highly discriminant items	Acute stroke patients	Clinician-observed	**Subscales (*n* _Item Total_ = 12):** General patient examination (6) Oral preparation phase (3) Oral phase (2) Pharyngeal phase (1)	4 and 5-level ordinal scales (different weighting)	**Range:** 20–100 (total score of all items); Cut-off score: ≥95 (start oral diet); <95 (non-oral diet) **Interpretation:** ↑ scores = ↓ impairment severity
**SFAM** Swallowing portion of the Functional Assessment Measure Hall [38]	Part of the Functional Assessment Measure (FAM) to address the patient’s functional status and document when assistance is required	First-time stroke patients	Clinician-observed	**Single scale (1):** Ability to safely eat a regular diet by mouth Need for assistance Fully independent Modified independence Supervised (Modified dependence) Minimal assist (Modified dependence Moderate assist (Modified dependence) Maximal assist (Dependent) Total assist	Single 7-level ordinal scale (1 = Total assistance; 7 = Complete independence)	**Range:** 1–7 **Interpretation:** ↓ scores = ↑ assistance required
**SPEAD** Swallowing Proficiency for Eating and Drinking Karsten, Hilgers [30]	A test which evaluates an individual’s (safe) swallowing capacity for eating as well as drinking	Head and neck cancer patients	Clinician-observed	**For three consistencies (IDDSI level 0, 3 and 7; *n* _Item Total_ = 15):** Total duration Grams swallowed Number of swallows Number of chews Coughing at any time during or directly after ingestion of the consistency **SPEAD-rate:** For each consistency; Mean SPEAD-rate	Numerical values; time	**Range:** N/A **Interpretation (item level):** ↑ scores = ↑ dysphagia severity **Interpretation (SPEAD-rate):** ↑ scores = ↑ swallowing capacity
**Swallowing Status** Moorhead, Johnson [39]	One of the five Nursing Outcome Classification (NOC) nursing outcomes that contain essential indicators to assess the entire swallowing process.	Stroke patients	Clinician-observed	**Single scale; Indicators (*n* _Item Total_ = 10):** Ability to bring food to the mouth Integrity of the chewing structures Ability to maintain oral content in the mouth Discomfort in swallowing the bolus Emptying of the oral cavity after swallowing the bolus Postural control of the head and neck relative to the body Cough Regurgitation Elevation of the larynx Aspiration respiratory	5-level ordinal scale (1 = worst health outcome; 5 = best health outcome)	**Range:** 10–50 **Interpretation:** ↑ scores = ↑ health outcomes
**TOMASS** Test of Masticating and Swallowing Solids Athukorala, Jones [40]	A quantitative assessment of solid bolus ingestion to evaluate oral preparation and oral phase of solids	Patients with Parkinson’s disease	Clinician-observed	During a **cracker swallow trial**, four measures are quantified **(*n* _Item Total_ = 4)**: Discrete bites per cracker Masticatory cycles per cracker Swallows per cracker Total time (in s)	Integer values; time	**Range:** N/A **Interpretation:** ↑ scores = ↑ dysphagia severity

Note. * ↑ or ↓, respectively, higher or lower; N/A = Not Applicable.

**Table 3 jcm-12-00721-t003:** Psychometrics reported for measures.

Measurement Tool	Reliability			Validity				
	Internal Consistency	Reliability ^†^	Measurement Error ^‡^	Content Validity	Criterion Validity	Construct Validity		
						Hypothesis Testing *	Cross-Cultural Validity **	Structural Validity ***
**ASHA-NOMS Dysphagia Scale** American Speech-Language-Hearing Association National Outcomes Measurement System [34] Dysphagia Scale Dungan, Gregorio [21]	NR	NR	NR	Development study NR Content validity study Relevance: NR Comprehensibility: NR Comprehensiveness: NR	N/A	***Aspect/Method:*** Convergent validity ***Results:*** Using Spearman correlations coefficient, the following correlations were shown with MASA at baseline (0.623) and discharge (0.832), with G-Codes at baseline (0.858) and at discharge (0.645), as well as with FOIS at baseline (0.919) and discharge (0.950) assessments [21]. ***Aspect/Method:*** ROC curve analysis ***Results:*** The FOIS and NOMS revealed AUROC coefficients of 0.808 and 0.849, respectively, indicating a similar utility in classifying dysphagia [21]. ***Aspect/Method:*** Known group validity ***Results:*** In the baseline evaluation, the ASHA NOMS score was significantly lower in the rostral group (2.2 ± 1.9) than in the caudal group (3.7 ± 2.4). At six-month evaluations, there was no significant difference between the two groups [41]. **IR** (see EDSQ [35])	N/A	NR
**DDS** Dysphagia Disorders Survey Sheppard, Hochman [28]	***Aspect/Method:*** Cronbach’s alpha ***Results*****:** Cronbach’s alpha used for RF (α = 0.89) and FSC (α = 0.89) subscales individually as well as together (α = 0.93) for the full DDS ratings [28].	***Aspect/Method:*** Inter-rater agreement ***Results:*** Kappa coefficient used for RF (κ = 0.63) and FSC (κ = 0.71) subscales individually as well as together (κ = 0.53) for the full DDS ratings [28].	NR	Development study NR Content validity study Relevance: NR Comprehensibility: NR Comprehensiveness: NR	N/A	***Aspect/Method:*** Convergent validity ***Results:*** Using Spearman correlations coefficient, a positive and significant strong correlation was found between the DMSS and RF (ρ = 0.88, *p* < 0.01) and FSC (ρ = 0.91, *p* < 0.01) subscales as well as the scores (ρ = 0.93, *p* < 0.01) for the full DDS [28]. ***Aspect/Method*:** Diagnostic accuracy ***Results*:** The RF subscale achieved a sensitivity ratio of 0.88 and a specificity ratio of 0.85 whilst these ratios for the FCS subscale were 0.94 and 0.87, respectively. Sensitivity and specificity ratios were 1.00 and 0.81 for the full DSS [28]. ***Aspect/Method*:** Predictive validity ***Results*****:** The NPV was 0.96 for the RF and 0.98 for the FCS, whilst PPV was 0.61 for the RF and 0.67 for the FCS. Negative and positive predictive values were 1.00 and 0.44 for the full DSS [28].	N/A	***Aspect/Method:*** Factor analysis (principal components analysis) ***Results:*** Results of the factor analysis consistently show that regardless of the sample or the sub-scale, that there is a single factor that accounts for approximately 50% of the total variance among the 15 valuables (seven items for the RF and eight items for the FSC) entered into the equation [28].
**DMSS** Dysphagia Management Staging Scale Sheppard, Hochman [28]	NR	NR	NR	Development study NR Content validity study Relevance: NR Comprehensibility: NR Comprehensiveness: NR	N/A	***Aspect/Method:*** Discriminative validity ***Results:*** Between two test sites, cross tabular analyses were conducted on the DMSS ratings comparing the two samples. Although the chi-square statistic was significant (χ^2^ = 47.84, 4 df, *p* < 0.01), the Mann–Whitney U statistic, which assesses ordinal differences between two categories was not (*p* = 0.41); the gamma statistic, which measures ordinality (γ = 0.06, *p* = 0.43) was also not significant [28]. IR (see DDS [28])	N/A	NR
**DSRS** Dysphagia Severity Rating Scale Everton, Benfield [26]	***Aspect/Method:*** Cronbach’s alpha ***Results:*** “Good” (Cronbach’s alpha α = 0.89 and α = 0.88) at baseline, varied between “Good” and “Excellent” (α = 0.88, α = 0.87 and α = 0.80–0.92) over the first two weeks, and “Excellent” (α = 0.92, α = 0.91 and α = 0.96) at 12 weeks [26].	***Aspect/Method:*** Inter-rater agreement ***Results:*** ICC was “Good” (ICC = 0.837, 95% CI 0.740, 0.900) for the Fluids subscale, “Excellent” (ICC = 0.985, 95% CI 0.974, 0.991) for the Diet subscale, “Excellent” (ICC = 0.952, 95% CI 0.921, 0.971) for the Supervision subscale, and “Excellent” (ICC = 0.955, 95% CI 0.925, 0.973) for the full DSRS score [26]. ***Aspect/Method:*** Intra-rater agreement ***Results:*** ICC was “Excellent” (ICC = 1.00, 95% CI 1.00, 1.00) for all three subscales and the full DSRS score [26].	NR	Development study The DSRS is a clinician-rated scale that was developed from the dysphagia outcome and severity scale (DOSS) [26]. Content validity study Relevance: Determined via survey responded to by 10 SLTs, all but two components of the DSRS subscales had “excellent” relevance (I-CVI > 0.90); The S-CVI was 0.84 (good) for the Fluids subscale, 0.84 (good) for the Diet subscale and 0.96 (excellent) for the Supervision subscale Comprehensibility: “Need for more detailed descriptors”—some respondents felt that more detail was needed to define terms, for example, “selected textures”, or that a description of bolus cohesiveness/food consistency should be included. Respondents also noted that some terms were subjective. Comprehensiveness: Responses from 10 UK-based SLTs indicated that the items were comprehensive for 30% of items in the Fluids subscale, 20% of items in the Solids subscale, and 60% of items in the Supervision subscale. Similarly, the wording was deemed clear for 50%, 30% and 80% of the items in the Fluids, Solids and Supervision subscales, respectively [26].	N/A	***Aspect/Method:*** Convergent validity ***Results:*** For the largest of the four trials reported on, Spearman’s rank correlation coefficients were determined for aspiration (PAS using VFS), swallowing ability (TOR-BSST and FOIS), disability (Barthel index), impairment (NIHSS) and dependency (modified Rankin Scale) at baseline and weeks 2 and 13. These values were rs = 0.488, rs = 0.387, and rs = 0.398 for VFS-PAS; rs = −0.167, rs = −0.459, and rs = −0.520 for TOR-BSST; rs = 0.020, rs = 0.301, and rs = 0.117 for NIHSS; rs = −0.279, rs = −0.517, and rs = −0.407 for the Barthel index, and rs = 0.179, rs = 0.382, and rs = 0.279 for the modified Rankin scale. For other trials, see paper [26].	N/A	NR
**EDACS** Eating and Drinking Ability Classification System Sellers, Mandy [33]	NR	***Aspect/Method:*** Inter-rater agreement ***Results:*** Kappa coefficient was κ = 0.866 (ICC = 0.867, 95% CI 0.813–0.906; *p* < 0.001) with 80.3% exact agreement between clinicians for EDACS as well as κ = 0.713 (ICC = 0.885, 95% CI 0.837–0.919; *p* < 0.001) with 88.0% exact agreement for level of assistance [42]. ***Aspect/Method:*** Inter-rater agreement ***Results:*** Kappa coefficient was *κ* = 0.884 (ICC = 0.717, 95% CI 0.538–0.830; *p* < 0.001) with 61.1% exact agreement between clinician and participant/ caregiver for EDACS as well as *κ* = 0.823 (ICC = 0.826, 95% CI 0.712–0.896; *p* < 0.001) with 79.6% exact agreement for level of assistance [42]. ***Aspect/Method:*** Inter-rater agreement ***Results:*** Absolute agreement between clinicians for EDACS was 78%, with kappa = 0.72 and ICC = 0.93 (95% CI 0.90–0.95) indicating substantial agreement. Absolute agreement between clinicians for degree of assistance was 87%, with Kappa = 0.80 and ICC = 0.92 (95% CI 0.88–0.94) indicating excellent agreement [33]. ***Aspect/Method:*** Inter-rater agreement ***Results:*** Absolute agreement between clinicians and parents for EDACS was 58% with kappa = 0.45 and ICC = 0.86. 95% (CI 0.76–0.92) indicating moderate agreement. Absolute agreement between clinicians and parents for level of assistance was 79% with kappa = 0.64 and ICC = 0.77. 95% (CI 0.62–0.87) indicating moderate to substantial agreement [33].	NR	Development study Development involved four distinct stages: (1) an original draft was constructed from the literature and clinical experience, (2) the draft was examined and revised using several iterations of a Nominal Group Process, (3) further examination and revision to the EDACS took place within two rounds of an online Delphi survey until agreement about the content was reached, and (4) the final stage assessed reliability between speech and language therapists and between speech and language therapists and parents [33]. Content validity study Relevance: NR Comprehensibility: NR Comprehensiveness: NR	N/A	***Aspect/Method:*** Convergent validity ***Results:*** Using Kendall’s tau-b, correlations were evaluated with FOIS (*K_τ_* = −0.346), SWAL-QOL (*K_τ_* = −0.389), total symptom score (*K_τ_* = −0.476), GMFCS (*K_τ_* = 0.140), and (*K_τ_* = 0.180) with MACS for EDACS as well as FOIS (*K_τ_* = −0.183), SWAL-QOL (*K_τ_* = −0.234), total symptom score (*K_τ_* = −0.263), GMFCS (*K_τ_* = 0.497), and (*K_τ_* = 0.584) for level of assistance [42]. ***Aspect/Method:*** Convergent validity ***Results:*** There was a significant positive correlation (Kendall’s tau = 0.69, *p* < 0.01) between EDACS level and level of assistance required to bring food and fluid to the mouth and a statistically significant but only moderate positive correlation (Kendall’s tau = 0.5, *p* < 0.01) between the EDACS and the GMFCS [33].	N/A	NR
**EDSQ** Easy Dysphagia Symptom Questionnaire Uhm, Kim [35]	***Aspect/Method:*** Cronbach’s alpha ***Results:*** Cronbach’s *α* coefficient was 0.785 [35]	NR	NR	Development study Following a review of existing questionnaires, the EDSQ was established by consensus of three physiatrists. We extracted 12 “yes/no” questions for dysphagia symptoms considering their easy applicability to older adults [35]. Content validity study Relevance: NR Comprehensibility: NR Comprehensiveness: NR	N/A	***Aspect/Method:*** Convergent validity ***Results:*** Showed significant correlations, using Spearman correlation analysis, with the MWST (*r* = −0.468, *p* = 0.001), the ASHA NOMS swallowing scale (*r* = −0.635, *p* < 0.001) and VDS (*r* = 0.449, *p* = 0.001) scales [35]. ***Aspect/Method:*** Diagnostic accuracy ***Results:*** According to the ROC curve analysis, the optimal cut-off score to maximize the sum of sensitivity and specificity was 5, with a sensitivity of 90.9% and a specificity of 67.5% [35].	N/A	NR
**FOIS** Functional Oral Intake Scale Crary, Carnaby Mann [23]	NR	***Aspect/Method:*** Inter-rater agreement ***Results:*** Kappa coefficient between paired judges ranged from *κ* = 0.86 to *κ* = 0.91, perfect agreement ranged from 85–95%, and Spearman’s rank correlations ranged from ρ = 0.98 to ρ = 0.99 [23].	NR	Development study Initially, the scale included 10 items. After pilot application in a tertiary care teaching hospital, unused items were omitted, and the remaining 7 items were retained for subsequent psychometric analysis. Levels 1 through 3 relate to varying degrees of nonoral feeding; levels 4 through 7 relate to varying degrees of oral feeding without nonoral supplementation. These latter scale levels consider both diet modifications and patient compensations, but all levels focus on what the patient consumes by mouth on a daily basis [23]. Content validity study Relevance: NR Comprehensibility: NR Comprehensiveness: NR	N/A	***Aspect/Method:*** Convergent validity ***Results:*** The FOIS score was significantly associated (χ², *p*, Cramer’s V correlation) with MRS (28.6, <0.001, 0.31), MBI (30.9, <0.001, 0.32), and the MASA (33.8, <0.001, 0.53) at admission; and then with MRS (64.9, <0.001, 0.49), MBI (64.6, <0.001, 0.49), and MASA (60.7, <0.001, 0.76) at 1-month post-stroke [23]. ***Aspect/Method:*** Hypothesis testing (reported as consensual validity) ***Results:*** Agreement with the predefined scale scores ranged from 81% to 98%. The Kendall concordance was 0.90 [23]. ***Aspect/Method:*** Convergent validity ***Results:*** The FOIS score was significantly associated (χ², *p*, Cramer’s V correlation) with both the presence of aspiration (30.17, <0.011, 0.40) and dysphagia (12.97, 0.011, 0.26) as well as dysphagia severity (56.48, <0.001, 0.54) but not significantly associated with aspiration severity [23]. ***Aspect/Method:*** Convergent validity ***Results:*** Using two-tailed Pearson correlations, between FOIS and PAS, r = −0.201 (*p* = 0.140) for semisolids and r = −0.218 (*p* = 0.110) for liquids. Between FOIS and pooling score, r = −0.355 (*p* = 0.008) for semisolids and r = −0.180 (*p* = 0.189) for liquids [43]. ***Aspect/Method:*** Diagnostic accuracy ***Results:*** When compared to PAS, for identifying dysphagia for liquids, FOIS had a sensitivity of 6.3% and a specificity of 94.9%, whilst for semisolids, these values were 6.1% and 95.5%, respectively. When compared with pooling score, sensitivity was 10% and specificity was 97.1% for liquids, whilst values were 13.6% and 100%, respectively, for semisolids [43]. ***Aspect/Method:*** Convergent validity ***Results:*** Using Spearman’s rank correlation rho at four time points, there was a weak and non-significant correlation (r = −0.20–0.13; *p* = 0.40–0.81) with PAS and a moderate significant correlation with EAT-10 (r = −0.53; *p* = 0.002) during treatment and (r = −0.56; *p* = 0.003) 3 months after treatment [44]. **IR** (see ASHA-NOMS [21], DSRS [26], EDACS [33], IDDSI-FDS [36], and SPEAD [30])	N/A	NR
**IDDSI-FDS** International Dysphagia Diet Standardisation Initiative Functional Diet Scale Steele, Namasivayam-MacDonald [36]	NR	***Aspect/Method:*** Inter-rater agreement ***Results:*** Using Kendall concordance, agreement was W = 0.873 overall and, across the successive quartile batches, was W = 0.88, W = 0.884, W = 0.896, and W = 0.819, respectively. Average ICCs per batch were 0.965, 0.966, 0.971, and 0.939, respectively, with 95% confidence interval boundaries of 0.872–0.976 [36].	NR	Development study NR Content validity study Relevance: Respondents indicated general agreement with the bracketed range concept (59% in favour) and 28% of respondents recommended that tolerance of consistencies between the bracketed boundaries on the IDDSI framework should not be assumed, but confirmed during assessment on a case-by-case basis. There was strong agreement (77%) that the IDDSI Functional Diet Scale score should reflect the main diet recommendation and not reflect therapeutic advancement [36]. Comprehensibility: NR Comprehensiveness: NR	N/A	***Aspect/Method:*** Convergent validity ***Results:*** Strong correlation (Spearman correlation: ρ = 0.84, *p* < 0.001) IDDSI-FDS with FOIS scores for case scenarios including diet texture recommendations [36].	N/A	NR
**MASA** Mann Assessment of Swallowing Ability Mann [27]	***Aspect/Method:*** Cronbach’s alpha ***Results:*** Cronbach’s alpha α = 0.9166 [27]	***Aspect/Method:*** Inter-rater agreement ***Results:*** Kappa coefficient was κ = 0.82 for dysphagia and κ = 0.75 for aspiration [27].	NR	Development study A panel of 15 content experts identified items that they felt should be included in a clinical assessment of dysphagia and provided feedback for minor modifications [27]. Content validity study Relevance: NR Comprehensibility: NR Comprehensiveness: NR	N/A	***Aspect/Method:*** ROC curve analysis ***Results:*** In ROC analysis for predicting the development of pneumonia, the cut-off value was 170.5 (sensitivity 0.70, specificity 0.83; AUC 0.82, 95% CI 0.78–0.87, *p* < 0.01) for MASA [45]. ***Aspect/Method:*** Diagnostic accuracy ***Results:*** As a predictor of aspiration, the MASA had a specificity of 69.9% and a sensitivity of 71.4%. According to previous cognitive assessment, patients were divided into subgroups based on cognitive function. Sensitivity ranged from 50.0% for mild and moderate to 90.9% for severely impaired patients, whilst specificity ranged from 38.1% for severe to 86.7% for mildly impaired patients [24]. ***Aspect/Method:*** Predictive validity ***Results:*** As a predictor of aspiration, the MASA has a positive predictive value of 64.3% and a negative predictive value of 76.3%. According to previous cognitive assessment, patients were divided into subgroups based on cognitive function. PPV ranged from 33.3% for mild to 69.8% for severe impairment, whilst NPV ranged from 63.6% for moderate to 92.9% for mild impairment [24]. ***Aspect/Method:*** Convergent validity ***Results:*** Using Spearman’s correlation coefficient, there was a significant positive correlation (r = 0.961, *p* < 0.01) with the mMASA [24]. ***Aspect/Method:*** Diagnostic accuracy ***Results:*** As a predictor of aspiration, the ORR and %TNS were calculated for the MASA. When using “probable” as the cut-off, specificity was 74.4% (95% CI 63.2, 83.6), sensitivity was 64.6% (95% CI 49.5, 77.8), PPV = 60.8 (95% CI 46.1, 74.1), and NPV = 77.3 (95% CI 67.8, 86.9) for MASA’s ORR. When using “moderate-severe” as the cut-off, specificity was 85.9% (95% CI 76.2, 92.7), sensitivity was 16.7% (95% CI 7.5, 30.2), PPV = 42.1 (95% CI 20.3, 66.5), and NPV = 62.6 (95% CI 52.7, 71.8) for MASA’s %TNS [46]. ***Aspect/Method:*** ROC curve analysis ***Results:*** ROC analysis resulted in an ROC area of 0.50 for the %TNS and 0.72 for the ORR [46]. **IR** (see ASHA-NOMS [21], FOIS [23], and MASA-C [29]).	N/A	NR
**MASA-C** Mann Assessment of Swallowing Ability—Cancer Carnaby and Crary [29]	***Aspect/Method:*** Cronbach’s alpha ***Results:*** Cronbach’s alpha α = 0.94 [29].	***Aspect/Method:*** Test-retest reliability ***Results:*** ICC = 0.96 at baseline and ICC = 0.92 at post-treatment [29]. ***Aspect/Method:*** Inter-rater agreement ***Results:*** ICC = 0.96 (95% CI 0.94–0.98) [29]. ***Aspect/Method:*** Intra-rater agreement ***Results:*** ICC = 0.94 (95% CI 0.91–0.97) [29].	NR	Development study Developers of the MASA-C identified specific items via literature review and a panel of 5 expert reviewers rated the potential new items, then revised items were selected on the basis of correlation, with items included on the basis of correlation for Cronbach’s alpha of α = 0.85 [29]. Content validity study Relevance: NR Comprehensibility: NR Comprehensiveness: NR	***Aspect/Method:*** ROC curve analysis ***Results:*** Presence of dysphagia (score ≤ 185): AUC = 0.95 (0.84–0.99; *p* < 0.0001); Se = 83%, Sp = 96%. Likewise, predictive values (PPV = 95%, NPV = 86%) were strong. Presence of aspiration (score ≤ 176): AUC = 0.90 (0.793–0.971; *p <* 0.0001), with Se, Sp, PPV and NPV not reported [29]. ***Aspect/Method:*** Criterion Validity ***Results:*** Using Spearman correlations coefficient, a moderately strong (r = 0.699) correlation was found with MASA [29].	***Aspect/Method:*** Convergent validity ***Results:*** Using Spearman correlations coefficient, strong correlation (r = 0.8295) with FOIS, a moderate (r = 0.488) correlation with FACT H&N, and a modest correlation (r = −0.3901) with VFE [29]. ***Aspect/Method:*** Predictive validity ***Results:*** Interpretation of the final model (log odds) revealed that for every 10-point rise in MASA-C score, the odds of achieving a favourable outcome posttreatment rose by 15.49 times compared to patients not improving their MASA-C score [29].	NR	***Aspect/Method:*** EFA ***Results:*** Four factors containing >4 items were retained producing a 23-item measure. Results showed that all items loaded significantly on their respective factors, ranging from 0.7 to 0.90 for acute effects, 0.54 to 0.73 for pharyngeal function, 0.65 to 0.8 for oral function, and 0.47 to 0.85 for cognitive–motor function [29].
**MISA** McGill Ingestive Skills Assessment Lambert, Gisel [37]	***Aspect/Method:*** Cronbach’s alpha ***Results:*** Cronbach’s alpha α = 0.86–0.91 across the 5 subscales [37].	***Aspect/Method:*** Inter-rater agreement ***Results:*** ICC = 0.85 (95% CI 0.78-.090) for total MISA score and ICC = 0.68 (95 % CI 0.55–0.78)–0.88 (95% CI 0.82–0.92) across the 5 subscales [30] ***Aspect/Method:*** Inter-rater agreement ***Results:*** Person’s correlation coefficient was used for each subscale, with r = 0.95 (80% exact agreement) for “Positioning”, r = 0.97 (93%) for “Textures”, r = 0.92 (67%) for “Self-feeding, r = 0.92 (45%) for Solid Ingestion, and r = 0.95 (75%) for Liquid Ingestion [37].	NR	Development study The development of the assessment began in 1996 with a review of the literature. To select items, a focus group of clinicians was assembled. At the end of item development, the assessment was named the McGill Ingestive Skills Assessment (MISA) and had 190 items in 7 scales. After a field test and item reduction, the adequacy of each of the items was examined. Items that had correlations >0.80 with at least one other item on the assessment were identified. Each pair of redundant items was inspected, and a judgment made whether to retain both items or eliminate one. If the items appeared to have a true redundancy, the item which was worded less clearly was eliminated [37]. Content validity study Relevance: NR Comprehensibility: NR Comprehensiveness: NR	N/A	***Aspect/Method:*** Convergent validity ***Results:*** Using Spearman correlations coefficient, a positive and significant moderate correlation (*ρ* = 0.45; *p <* 0.05) was found with the Functional Independence Measure and a positive yet weak correlation (ρ = 0.25, *p* < 0.05) was found with 3MS [32]. ***Aspect/Method:*** Contrasted groups validity ***Results:*** Using Wilcoxon Rank-Sum tests, *p* = 0.11 (*p* > 0.05) for participants taking psychoactive medication, *p* = 0.28 (*p* > 0.05) for participants with decubiti, and *p* = 0.01 (*p* < 0.05) for participants who wear dentures during meals [32]. ***Aspect/Method:*** Predictive validity ***Results:*** Survival analyses revealed that the MISA scores are predictive of death using a Cox proportional hazards model (hazard ratio = 0.960; 95% CI 0.940, 0.980), and of time to pulmonary infection using a flexible model. Scores on the Solid Ingestion and Self-feeding scales are predictive of death using the Cox model, and the Texture Management scale is predictive of death using the flexible model [47]. ***Aspect/Method:*** Contrasted groups validity ***Results:*** Using Spearman correlations, The MISA score correlated moderately with age (r = −0.58, *p* < 0.001) but was low with gender (r = −0.34, *p* < 0.02). Both were negative and only age was significant. The relationship with stroke severity (discharge destination) was significant (*H* = 12.7, df = 3, *p <* 0.005). Dysphagia status was highly significant (*p <* 0.0001), but location of lesion was not (*p <* 0.01). Correlations between the MISA score and first or repeated stroke, and between MISA score and location of lesion were low, negative, and non-significant (*r* = −0.07, *p <* 0.67 and *r* = −0.14, *p <* 0.35), respectively. Low and non-significant correlations were obtained between the MISA score and type of stroke (*r* = 0.06, *p <* 0.7) [48].	N/A	NR
**M-MASA/** **mMASA** Modified Mann Assessment of Swallowing Ability Antonios, Carnaby-Mann [22]	***Aspect/Method:*** Cronbach’s alpha ***Results:*** Cronbach’s alpha α = 0.94 [22].	***Aspect/Method:*** Inter-rater agreement ***Results:*** Kappa coefficient of *κ* =0.76 (SE = 0.082) between neurologists [22].	NR	Development study: Original MASA data were statistically reviewed to identify potential screening items. Screening items were selected on the basis of correlations with total (the correlation with the total score had to be at least 0.4), and each item’s individual Cronbach alpha (≥0.85). In addition, the items were considered with regard to the familiarity and use of each potential item within currently administered clinical neurologic assessments [22]. Content validity study Relevance: NR Comprehensibility: NR Comprehensiveness: NR	***Aspect/Method*:** ROC curve analysis using optimal cut-off point ***Results*:** The optimal cut point on the mMASA to identify dysphagia was 94 of 100 possible points. Using this cut-off point, AUC = 0.93 (95% CI: 0.89–0.97) for the first rater and AUC = 0.94 (95% CI: 0.87–0.96) for the second rater [22].	***Aspect/Method:*** Convergent validity ***Results:*** Using Spearman’s correlation coefficient, there was a moderate negative (r = −0.349, *p* = 0.044) correlation with PAS. Additionally, with cognitive function tests, significant correlations were shown, with ρ = 0.564 (*p* < 0.05) for MMSE, ρ = 0.641 (*p* < 0.05) for MoCA, and ρ = −0.676 (*p* < 0.05) for CDR [49]. ***Aspect/Method:*** Diagnostic accuracy ***Results:*** Based on results from the ROC analysis of the two raters, detection of dysphagia was high (sensitivity = 92% and 87%; specificity = 86.3% and 84.2%), whilst PPV was 79.4% and 75.8% and NPV was 95.3% and 92% [22].	NR	NR
**SFAM** Swallowing portion of the Functional Assessment Measure Hall [38]	NR	***Aspect/Method:*** Inter-rater agreement ***Results:*** ICC = 0.975 (*p*≤ 0.01) at admission and ICC = 0.964, (*p*≤ 0.01) at discharge as well as Spearman rho correlations of ρ = 0.899 (*p* < 0.01) at admission and ρ = 0.863 (*p* < 0.01) at discharge [50].	NR	Development study NR Content validity study Relevance: NR Comprehensibility: NR Comprehensiveness: NR	N/A	***Aspect/Method:*** Convergent validity ***Results:*** Spearman rho correlations were performed on the SFAM and FOIS, and a strong significant relationship was found (ρ = 0.926, *p* < 0.01) at admission and (ρ = 0.706, *p* < 0.01) at discharge [50]. ***Aspect/Method:*** Convergent validity ***Results:*** Strong correlations (r = 0.779, *p* ≤ 0.001) with the food texture ratings at admission, and the SFAM levels and the liquid consistency ratings (r = 0.762, *p* ≤ 0.001) at admission. Moderately strong correlations (r = 0.673, *p* ≤ 0.001) were apparent between the SFAM levels and the food texture ratings at discharge as well as with the SFAM levels and the liquid consistency ratings (0.567, *p* ≤ 0.001) at discharge [51]. ***Aspect/Method:*** Predictive validity ***Results:*** When predicting discharge for age, 72% of younger (50 years old and younger) patients reached a SFAM Level 5, 6, or 7 (mild to no dysphagia) as compared to 51% of older patients. 59% of younger patients had a length of stay of 14 days or less as compared to 27% of the older patients. When predicting discharge for patients with a cognitive FIM score of 14 or lower, 82% had severe dysphagia (SFAM score of 1 or 2) as compared to 35% that had moderate dysphagia (SFAM score of 3 or 4). 61% had a length of stay of 15 days or more as compared to 39% who had a length of stay of 14 days or less [52].	N/A	NR
**SPEAD** Swallowing Proficiency for Eating and Drinking Karsten, Hilgers [30]	NR	***Aspect/Method:*** Test-retest reliability ***Results:*** ICC = 0.90 (0.86–0.94), 0.88 (0.83–0.92), 0.89 (0.83–0.93) of duration and ICC = 0.84 (0.77–0.90), 0.68 (0.56–0.78), 0.60 (0.46–0.73) of number of swallows for thin, thick and solid consistencies, respectively. ICC = 0.89 (0.83–0.93) for number of chews [30]. ***Aspect/Method:*** Intra-rater agreement ***Results:*** ICC = 1.00 (1.00–1.00), 0.98 (0.97–0.99), 0.98 (0.96–0.98) of duration and ICC = 0.99 (0.99–1.00), 0.96 (0.95–0.97), 0.96 (0.94–0.97) of number of swallows for thin, thick and solid consistencies, respectively. ICC = 1.00 (0.99–1.00) for number of chews [30]. ***Aspect/method:*** Inter-rater agreement ***Results:*** ICC = 0.98 (0.97–0.99), 0.97 (0.96–0.98), 0.95 (0.93–0.97) of duration and ICC = 0.93 (0.90–0.95), 0.74 (0.65–0.81), 0.75 (0.65–0.82) of number of swallows for thin, thick and solid consistencies, respectively. ICC = 0.98 (0.98–0.99) for number of chews [30].	NR	Development study NR Content validity study Relevance: NR Comprehensibility: NR Comprehensiveness: NR	N/A	***Aspect/Method:*** Convergent validity ***Results:*** Spearman correlations coefficient was used with SPEAD-rate and subjective swallowing outcomes, with ρ = 0.71 (*p* < 0.001) for self-rated percentage eating and drinking speed, ρ = 0.72 (*p* < 0.001) for self-rated percentage swallow function, ρ = −0.68 (*p* < 0.001) for SWAL-QOL total score, and ρ = −0.70 for degree of dysphagia by SLP. Similarly, correlations were found with SPEAD-rate and objective swallowing outcomes, with *ρ* = 0.70 (*p <* 0.001) for FOIS, ρ = −0.51 (*p* = 0.001) for DIGEST grade, ρ = −0.50 (*p* = 0.001) for aspiration on VFS, and ρ = 0.49 (*p* < 0.001) for maximal mouth opening [30]. ***Aspect/Method:*** Divergent validity ***Results:*** Correlations of the SPEAD-rate with participant-reported dyspnoea, pain and fatigue were weak (ρ between 0.25 and 0.28), again using Spearman correlations coefficient [30]. ***Aspect/Method:*** Discriminant validity ***Results:*** As hypothesized, patients had a median SPEAD-rate of 2 g/s (range 0–10), compared to 6 g/s (range 2–11) for healthy participants corresponding to a large effect size of 0.56. When dividing participants into four groups based on degree of dysphagia rated by the SLP (no, mild, moderate and severe, with the healthy participants rated as no), SPEAD-rate decreases (*p* < 0.001) with increasing degree of dysphagia [30]. ***Aspect/Method:*** Diagnostic accuracy ***Results:*** When using the SPEAD-rate to discriminate between patients and healthy participants, the area under the ROC-curve was 0.82, with a cut-off value for optimal sensitivity and specificity ratio of 4.2 g/s (sensitivity 80% and specificity 79%). When using the SPEAD-rate to determine aspiration, the area under the ROC-curve was 0.79, with an optimal cut-off value of 1.2 g/s, giving 100% sensitivity and 57% specificity [30].	N/A	NR
**Swallowing Status** Moorhead, Johnson [39]	***Aspect/Method:*** Cronbach’s alpha ***Results:*** Cronbach’s α for the overall scale = 0.954; values for when individual items were deleted ranged from α = 0.945 to α = 0.956 [25]. ***Aspect/Method:*** ***Results:*** Person reliability estimate = 0.905; indicating good internal consistency [53].	***Aspect/Method:*** Test-retest reliability ***Results:*** ICC reported per indicator with values ranging from ICC = 0.571 (0.258–0.776) to ICC = 1.00 (1.00–1.00) for initial evaluation and from ICC = 0.727 (0.410–0.874) to ICC = 1.00 (1.00–1.00) after 72 h [25].	NR	Development study An integrative review was performed for the Conceptual Analysis, which enabled finding papers addressing this topic, in addition to dissertations, theses and books. The NOC indicators were revised, and conceptual and operational definitions were developed for each indicator. Additionally, for each magnitude, that is, for each of the five points on the Likert scale, an operational definition was established to help nurses during assessments [54]. Content validity study ICCs were used to determine agreement between nurses both with definitions, ranging from ICC = 0.899 (95% CI 0.848–0.934) to ICC = 1.00 (95% CI 1.00–1.00) across all indicators, and without definitions, ranging from ICC = −0.071 (95% CI −0.260–0.131) to ICC = 0.626 (95% CI 0.368–0.775) across all indicators [54]. Relevance: A panel of 11 judges examined the relevance and clarity of each indicator, with one indicator not reaching the CVI cut-off point of 0.80. This indicator was retained due to its importance to clinical practice according to the literature [54]. Comprehensibility: Regarding clarity, each indicator and respective definition was examined according to the following: −1 (inappropriate definition/indicator), 0 (somewhat appropriate definition/indicator), and +1 (appropriate definition/indicator). The judges had the liberty to suggest changes concerning the names of the indicators, on their grouping or exclusion [54]. Comprehensiveness: NR	N/A	***Aspect/Method:*** IRT (2PPC Model) ***Results:*** The parametric bootstrap approximation to Pearson chi-squared goodness-of-fit measure found that values obtained in the sample are similar to those obtained from the model (*p* = 0.510). The fit on the two-way margins based on 2PPC model did not present discrepancies in percentages of adjustment among indicators (chi-square residuals < 3.5), denoting similarity between observed frequencies in sample and expected frequencies from the model. These results show good fit to the model and unidimensionality of the scale [25].	***Aspect/Method:*** Conducted Differential Item functioning (DIF) analysis for gender, age, type of stroke, and stroke severity [25]. ***Results:*** The measure did not show DIF for gender, age, type of stroke, and severity of stroke, indicating that these characteristics did not affect the final Swallowing Status outcome [25].	***Aspect/Method:*** Rasch analysis ***Results:*** The results showed good fit to the model and that the measure is unidimensional [25]. Analysis of standardized residuals in PCA indicates that the Rasch dimension explained 67.7%of data variance. It was slightly above guidelines for assessing unidimensionality via PCA (50%). The largest secondary dimension (the first contrast in the residuals) explained 8.4%. Many indicators presented disordered categorical response thresholds, with overlapping categories, which suggested that the scale of responses (5 points) was not appropriate and contributed to inadequate fit of items Changing from 5 points to 3 points showed the best fit for the items and people. Item difficulty for the NOC with 3 points ranged from −1.36 to 2.40 and infit statistics from 0.75 to 1.34 indicated good fitness [53].
**TOMASS** Test of Masticating and Swallowing Solids Athukorala, Jones [40]	***Aspect/Method:*** Cronbach’s alpha ***Results:*** Cronbach’s alpha ranged from α = 0.71 to α = 0.82 [55] ***Aspect/Method:*** Test-retest reliability ***Results:*** Cronbach’s alpha ranged from α = 0.94 to α = 0.99 [31].	***Aspect/Method:*** Inter-rater agreement ***Results:*** Values ranged from ICC = 0.96 (95% CI 0.927–0.979) to ICC = 1.0 across items for the clinical group and from ICC = 0.97 (95% CI 0.950–0.986) to ICC = 0.99 (95% CI 0.995–0.998) for the control group [56]. ***Aspect/Method:*** Intra-rater agreement ***Results:*** Values ranged from ICC = 0.97 (95% CI 0.954–0.987) to ICC = 1.0 across items for the clinical group and from ICC = 0.99 (95% CI 0.984–0.995) to ICC = 1.0 for the control group [56]. ***Aspect/Method:*** Inter-rater agreement ***Results:*** ICC > 0.95 for the number of masticatory cycles and time taken. The ICC = 0.73 for interrater reliability of the number of swallows recorded by instrumental assessment [31]. ***Aspect/Method:*** Test-retest reliability ***Results:*** ICC values ranged from ICC = 0.83 to ICC = 0.98 [31]. ***Aspect/Method:*** Inter-rater agreement ***Results:*** Median ICCs were reported with 95% confidence intervals for each outcome measure: ICC = 0.92 (0.75–0.91) for number of bites, ICC = 0.97 (0.92–0.96) for masticatory cycles, ICC = 1.00 (0.99–1.00) for total time, and ICC = 0.58 (0.51–0.67) for number of swallows, with 90% (93%-97%) exact agreement for signs of aspiration [57]. ***Aspect/Method:*** Intra-rater agreement ***Results:*** Median ICCs were reported with 95% confidence intervals for each outcome measure. For immediate agreement, ICC = 1.00 (0.99–1.00) for number of bites, ICC = 1.00 (0.98–1.00) for masticatory cycles, ICC = 1.00 (0.95–1.00) for total time, and ICC = 0.90 (0.87–0.97) for number of swallows, with 100% (96%-100%) exact agreement for signs of aspiration. For delayed agreement, ICC = 0.99 (0.91–1.00) for number of bites, ICC = 0.98 (0.96–0.99) for masticatory cycles, ICC = 1.00 (0.99–1.00) for total time, and ICC = 0.81 (0.71–0.87) for number of swallows, with 100% (94%-100%) exact agreement for signs of aspiration [57,58].	NR	Development study TOMASS was developed from the timed water swallow test to assess the swallowing rate of solids. Within the initial small sample, surface electromyography (EMG) measures derived from the masseter muscles were highly correlated with visual observation of chewing cycles, with the average Pearson correlation coefficient across four measurement sessions at *r* = 0.93, *p* < 0.05 [40]. Content validity study Relevance: NR Comprehensibility: NR Comprehensiveness: NR	N/A	***Aspect/Method:*** Convergent validity ***Results:*** Using Spearman correlations coefficient, a positive and significant moderate correlation between ‘number of swallows per cracker’ and mealtime duration (*r* = 0.49; *p <* 0.002; 95% CI = 0.20–0.70) and a positive and significant moderate correlation between ‘total time’ and mealtime duration (*r* = 0.41; *p <* 0.011; 95% CI= 0.10–0.65) were found. Bland-Altman plot between ‘number of swallows per cracker’ and ‘number of white-outs’ observed during FEES was −0.02 with 95% CI = −1.7 to 1 [56]. ***Aspect/Method:*** Known group validity ***Results:*** To analyse known-group validity, a Mann–Whitney U-test was used to compare the clinical and the control group, with U = 645 (*p* = 0.224) for discrete bites per cracker, U = 527.5 (*p* < 0.05) for masticatory cycles per cracker, U = 520.5 (*p* < 0.05) for swallows per cracker, and U = 457.5 (*p* < 0.05) for total time [56]. ***Aspect/Method:*** Hypothesis testing ***Results:*** The ICC value between objective and behavioural measures of the number of masticatory cycles was 0.99 with a 95% confidence interval from 0.98 to 0.99. For number of swallows, the ICC was 0.85 with a 95% confidence interval from 0.79 to 0.90. The ICC for time was 0.99 with a 95% confidence interval from 0.91 to 1.0 [31]. ***Aspect/Method:*** Contrasted groups validity ***Results:*** The effects of sex were significant across all variables (discrete bites, masticatory cycles and swallows per cracker, total time to ingest, masticatory cycles per bolus, and swallows per bolus) with the exception of the derived measures of average time per masticatory cycle and average time per swallow [31].	N/A	NR

Note. %TNS = Percentage Total Numeric Score; 3MS = Modified Mini-Mental State Examination; CDR = Clinical Dementia Ration; CI = Confidence Interval; EFA = Eploratory Factor Analysis; AUC = Area Under the Curve; FACT H&N = Functional Assessment of Cancer Therapy—Head and Neck; GMFCS = Gross Motor Function Classification System; ICC = Intra-Class Coefficient; I-CVI = Item-Content Validity Index; IR = Indirectly Reported elsewhere; MBI = Modified Barthel Index; MMSE = Mini-Mental State Examination; MoCA = Montreal Cognitive Assessment; MRS = Modified Rankin Scale; N/A = Not Applicable; NPV = Negative Predictive Value; NR = Not Reported; ORR = Ordinal Risk Rating; PPV = Positive Predictive Value; ROC = Receiver Operator Characteristic Curves; S-CVI = Scale-Content Validity Index; Se = Sensitivity; Sp = Specificity; VFE = Videofluoroscopic Swallowing Examination; ^†^ Including test-retest, intra, inter, ICC or Kappa; ^‡^ SDC or LoA MIC; * Hypothesis about other instruments and relation; ** Including differential item functioning DIF, Measurement invariance, or IRT—Rasch Analysis; *** Including CTT—Factor analysis or IRT -Rasch analysis.

**Table 4 jcm-12-00721-t004:** Summary of measurement properties evaluated for each measure.

Measurement Instrument	RELIABILITY			VALIDITY				
	Internal Consistency	Reliability †	Measurement Error ‡	Content Validity	Criterion Validity	Construct Validity		
						Hypothesis Testing *	Cross-Cultural Validity **	Structural Validity ***
**ASHA-NOMS DS**	NR	NR	NR	NR	N/A	Yes ◊	N/A	NR
**DDS**	Yes	Yes	NR	NR	N/A	Yes	N/A	Yes
**DMSS**	NR	NR	NR	NR	N/A	Yes ◊	N/A	NR
**DSRS**	Yes	Yes	NR	Yes	N/A	Yes	N/A	NR
**EDACS**	NR	Yes	NR	Yes	N/A	Yes	N/A	NR
**EDSQ**	Yes	NR	NR	Yes	N/A	Yes	N/A	NR
**FOIS**	NR	Yes	NR	Yes	N/A	Yes ◊	N/A	NR
**IDDSI-FDS**	NR	Yes	NR	Yes	N/A	Yes	N/A	NR
**MASA**	Yes	Yes	NR	Yes	N/A	Yes ◊	N/A	NR
**MASA-C**	Yes	Yes	NR	Yes	Yes	Yes	NR	Yes
**MISA**	Yes	Yes	NR	Yes	N/A	Yes	N/A	NR
**M-MASA/mMASA**	Yes	Yes	NR	Yes	Yes	Yes	NR	NR
**SFAM**	NR	Yes	NR	NR	N/A	Yes	N/A	NR
**SPEAD**	NR	Yes	NR	NR	N/A	Yes	N/A	NR
**Swallowing Status**	Yes	Yes	NR	Yes	N/A	Yes	Yes	Yes
**TOMASS**	Yes	Yes	NR	Yes	N/A	Yes	N/A	NR

Note. Legend background colours: Green = Yes; Yellow = Not Applicable; White = Not reported. ASHA-NOMS DS = ASHA-NOMS Dysphagia Scale; DDS = Dysphagia Disorders Survey; DMSS = Dysphagia Management Staging Scale; DSRS = Dysphagia Severity Rating Scale; EDACS = Eating and Drinking Ability Classification System; EDSQ = Easy Dysphagia Symptom Questionnaire; FOIS = Functional Oral Intake Scale; IDDSI-FDS = International Dysphagia Diet Standardisation Initiative Functional Diet Scale; MASA = Mann Assessment of Swallowing Ability; MASA-C = Mann Assessment of Wallowing Ability-Cancer; MISA = McGill Ingestive Skills Assessment; M-MASA/mMASA = Modified Mann Assessment of Swallowing Ability; SPEAD = Swallowing Proficiency for Eating and Drinking; TOMASS = Test of Masticating and Swallowing Solids. NA = not applicable; NR = not reported; ◊ Indirectly reported (i.e., not main focus of study); ^†^ Including test-retest, intra, inter, intraclass correlation coefficient or Kappa; ^‡^ smallest detectable change or limits of agreement or minimal important change; * Hypothesis about relation between included measure and other instrument(s); ** Including differential item functioning, Measurement invariance, or item response theory (Rasch Analysis); *** Including classic test theory (Factor analysis) or item response theory (Rasch analysis).

## Data Availability

No new data were created or analysed in this study. Data sharing is not applicable to this article.

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
