# Peer review of "Reliability and Validity of Non-Instrumental Clinical Assessments for Adults with Oropharyngeal Dysphagia: A Systematic Review"

_jcm, 2023, doi:10.3390/jcm12020721_

Round 1

Reviewer 1 Report

Dear Authors,

thank you for the opportunity to review this interesting research paper, however I have some important comments and serious concern for the publication of tour manuscript

1) The protocol of the systematic review, was it registered on PROSPERO, Campbell Collaboration or similar databases? Please provide the registration number.

2) Similar review was already published in 2022. Authors  already performed a SR on outcome measures and screening tool. Please see 

https://www.mdpi.com/2039-4403/12/2/25

I suggest to explain why you choose to perform so similar study with similar results. This represent a critical issue

3) Methodological Quality: Please provide more information on QualSyst. This can help readers to better understan how this approach works to verify methodological quality. Why to not choose Cosmin riso of Biasi check list?

4) I suggest to better describe (shortly) different tools

5) tables shouldbe used to synthetize, Table 3 is non useful for me. I suggest to reduce information, reporting tables with values  for Cronbach, ICC etc..

6) table 4 why choose to not use Cosmin?

7) in discussion you stated to follow cosmin-Based methodology, but you did not use Cosmin tools. Furthermore as future perspective you suggest to perform a New SR following cosmin aproach. This has no sense for me, because you have all the information/data to do this (and I suggest to do this). Furthermore, performing a new SR we will have the third SR on the same topic.

Reviewer 2 Report

Review report 20-11-2022

 A brief summary

Article Reliability and Validity of Non-instrumental Clinical Assessments for Adults with Oropharyngeal Dysphagia: A systematic review

Relevance

Oropharyngeal dysphagia can occur in adults and in older patients with Parkinson's disease, stroke, head and neck cancer patients, motor neurone disease, and in patients with dementia or xerostomia. The main consequences of OD are malnutrition, dehydration, and aspiration pneumonia. Timely diagnosis and treatment of oropharyngeal dysphagia are essential to ensure adequate nutrition and hydration in these patients. However, there are many tools for screening and clinical assessment of OD.

Some of them are designed for patients with a particular disorder, such as stroke or Parkinson‘s disease, whereas others are suited for the general population of older people. In addition, some tools are only suitable for simple screening, whereas others can be used to assess the patient‘s condition.

Therefore, a relevant question is which of these tools is the most reliable and suitable for use.

This review is a first step towards optimising non-instrumental clinical assessment of OD.

I would like to congratulate the authors for having completed this comprehensive study in search of the best non-instrumental clinical assessment tool for adult OD.

Comments

-

Rating the Manuscript

The topic of this study is relevant and important, especially for geriatricians and neurologists.

The manuscript clear, relevant for the field and presented in a well-structured manner.

The article contains all of the necessary components, and the methodology is clearly explained.

The figures/tables/schemes are appropriate.

The paper can in principle be accepted in present form.

The limitations of the study were provided.

Reviewer 3 Report

Dear authors,

the article is very well-written, and the systematic review has been carefully implemented and described. Thank you for giving me an opportunity to review this paper. While reading the article, a few questions came to mind that I would like the authors to comment on.

#1 A total of 14 references are used in the introduction chapter, and 11 of these are previous articles at least by one of the authors of this current manuscript. It would be nice if authors would justify using their own articles or consider broadening the perspective by also using articles written by other researchers.

#2 In my opinion, in chapter Systematic Literature Search was somewhat unnecessary repetition with Figure 1. Could the text of the chapter be capsulized to avoid repetition? For example, the sentence “These measures were excluded for the following reasons: 43 were screening tools, 34 were self-report tools, 16 measures were not developed in English, 13 as less than 50% of the assessment items were related to swallowing; 7 were instrumental assessments, 6 were related to oesophageal problems, and 6 were not clinical assessments at all.” could be removed. In addition, there is repetition in the last two sentences of this chapter.

#3 On first reading I found it difficult to understand what the up and down arrows in the last column of Table 2 mean. I guess I got it on second reading, but I wonder if there is a way to clarify this.

#4 It would be nice if authors would unify marking methods in Table 3, that is whether there is a space between the characters or not, for example ICC=0.885 vs. ICC = 0.717.

#5 Tables 2 and 3 are long and especially Table 3 (33 pages!) is difficult to read because the text continues to the second page, and you need check which measurement tools information is being read by returning to the previous page. It would be more comfortable for the reader if the table would be laid so that the information of one measurement fits on one page.

#6 While reading the chapter “Conceptual Mapping of Non-instrumental Clinical Measures”, I thought that this paragraph describes how the conceptual mapping has been done, so should it belong to the method chapter instead of the results chapter. I leave that decision to the authors and editors. In addition, I suggest that the authors would clarify what is “OD non-instrumental clinical assessment theory and definitions” to which they refer in their analysis. It is a bit unclear to me that was analysis done in this study theory-based thematic analysis or inductive analysis? This matter would be good to clarify too.

#7 Please, check the consistency of the referencing technique in the reference list according to the journal's instructions. For example,

Nordio, S.; et al., The correlation between pharyngeal residue, penetration/aspiration and nutritional modality: A cross-sectional 214 study in patients with neurogenic dysphagia. Acta Otorhinolaryngol Ital, 2020. 40(1): P. 38-43.

Ishii, M.; et al., Higher Activity and Quality of Life Correlates with Swallowing Function in Older Adults with Low Activities 216 of Daily Living. Gerontology, 2021: P. 1-9.

Round 2

Reviewer 1 Report

Dear Authors,

thank you for answer to some of my points. However I am continuing to have some concern.

PROSPERO OR OTHER DATABASE: You wrote:  Due to a grant commitment, we were required to finalise our review within a short timeframe (few months). Since registration may take up to several weeks or longer, we decided not to register our review in order to meet the grant requirements.

I do not believe that 'time constraints' can be a valid motivation. As registration is required for RCTs, the same must be applied for SR. I suggest at the very least submitting the protocol to PROSPERO (or other) and, once completed, writing the provisional code in the article. Specifyin as limit.

COSMIN

For what concern COSMIN, I suggest to see https://cosmin.nl/wp-content/uploads/COSMIN-syst-review-for-PROMs-manual_version-1_feb-2018.pdf. Please consider for the future that now an international panel of expert is working to develop PRISMA-COSMIN guidelines (https://systematicreviewsjournal.biomedcentral.com/articles/10.1186/s13643-022-01994-5) 

It is not true that you have to publish three different papers. Please see the following research : https://link.springer.com/article/10.1007/s40279-020-01268-x
(Authors of the research paper have developped COSMIN..)

I think your work is really interesting, but methodology must be different. We cannot accept to produce an overlapping of research paper. I suggest to use COSMIN methodology in this research paper.
